# Telerehabilitation of Post-Stroke Patients as a Therapeutic Solution in the Era of the Covid-19 Pandemic

**DOI:** 10.3390/healthcare9060654

**Published:** 2021-05-31

**Authors:** Paulina Magdalena Ostrowska, Maciej Śliwiński, Rafał Studnicki, Rita Hansdorfer-Korzon

**Affiliations:** Department of Physiotherapy, Medical University of Gdańsk, 7 Dębinki Street, 80-211 Gdańsk, Poland; sliwinskim@gumed.edu.pl (M.Ś.); rafal.studnicki@gumed.edu.pl (R.S.); rita.hansdorfer-korzon@gumed.edu.pl (R.H.-K.)

**Keywords:** Covid-19, telerehabilitation, post-stroke rehabilitation, virtual reality, stroke

## Abstract

(1) *Background*: Due to the pandemic caused by the SARS-CoV-2 virus, rehabilitation centres have become less available for neurological patients. This is the result of efforts to physically distance society, to try to slow the spread of the pathogen. Health care facilities were mainly restricted to urgent cases, while most physiotherapy treatments, mainly for patients with chronic conditions, were suspended. Some countries have seen a reduction in acute stroke hospital admissions of from 50% to 80%. One solution to the above problem is the use of telerehabilitation in the home environment as an alternative to inpatient rehabilitation. (2) *Aim of the study*: The purpose of this review is to analyse the benefits and limitations of teletherapy in relation to the functional condition of post-stroke patients. (3) *Methods*: Selected publications from 2019 to 2021 on the telerehabilitation of stroke patients were reviewed. The review was based on the Preferred Reporting Items for Systematic Reviews and Meta-Analysis (PRISMA) checklist. (4) *Results*: Studies have proven that teletherapy significantly improves the functional condition of post-stroke patients, resulting in improved quality of life and faster return to independence (while maintaining maximum possible precautions related to the SARS-CoV-2 virus pandemic). (5) *Conclusions*: Analysis of the study results showed comparable effectiveness of rehabilitation in the tele system to inpatient therapy. However, it should be emphasised that patients undergoing telerehabilitation must meet strict conditions to be eligible for this type of treatment program. However, the strength of the evidence itself supporting the effectiveness of this method ranks low due to the limited number of randomised control trials (RCT), small number of participants, and heterogeneous trials.

## 1. Introduction

Of the 15 million people worldwide who suffer a stroke each year, 5 million are permanently disabled and require ongoing care [1]. It is important to ensure access to rehabilitation within the first 3 months after stroke, as up to 80% of hospital readmissions are related to health habits and lack of appropriate therapy and post-stroke care [2]. Patients are generally discharged from hospital rehabilitation units 8–10 weeks after a stroke incident [3]. The functional condition of patients is characterised by significant motor deficits, and they require assistance with basic daily living activities (e.g., toilet use or eating) [4]. This is a period of increased post-damage (compensatory) plasticity in which major reorganizational changes occur in the brain. Therefore, it is critical to provide intensive, continuous, highly repetitive, task-oriented therapy to patients during this time. However, the reduction in the availability of inpatient post-stroke rehabilitation due to the SARS-CoV-2 virus pandemic has many serious consequences [5]. Above all, it puts patients at risk of acquiring pathological motor patterns and exacerbation of disability. Dhand’s research shows that social isolation, which negatively affects the functional and emotional state of post-stroke patients, is an individual risk factor for stroke recurrence [6]. However, this single report should not be valid with respect to the entire population of stroke survivors. A potential solution to the problem of the increasing demand for rehabilitation services in resource-limited conditions is a teletherapy system, which involves the provision of rehabilitation services at home without direct contact between the therapist and the patient through communication technologies (mainly auditory, visual, tactile) [7,8]. Contact with the physiotherapist can be synchronous (communication with the therapist occurs over video chats during actual therapy time) or asynchronous (communication with the therapist occurs occasionally, outside of therapy time). Physiotherapists verify the progress, validate the correctness of motor tasks, and update therapy plans based on video conferencing feedback, system usage data, and game scores. The telerehabilitation system also uses virtual-reality-based therapy, which uses computer software to track users’ movements and allows them to interact with a game or scenario presented on a television screen [9,10].

## 2. Materials and Methods

The review was based on research material obtained from PubMed, Directory of Open Access Journals, and Science Direct databases. The multi-search engine of the Main Library of the Medical University of Gdańsk was used to search for publications (search term: “telerehabilitation after stroke”). Study dates ranged from 2019 to 2021. Seventy-one records (PubMed—48, Directory of Open Access Journals—20, Science Direct—3) were identified and narrowed to the topics of “telerehabilitation” + “stroke”. The results were restricted to full text only. After removing duplicates, the authors independently analysed the titles and abstracts of 28 papers. Ten publications were selected, evaluating the effectiveness of teletherapy with respect to various aspects of the functional status: motor skills of the directly involved upper limb, balance, cardiorespiratory fitness, cognitive function, and the level of acceptance and feasibility of teletherapy by its users at home. The selection was based on inclusion criteria: (1) study involving post-stroke patients, (2) describing therapeutic interventions using telerehabilitation, (3) determining the effect of teletherapy on the functional condition of post-stroke patients, (4) evaluating the effectiveness of teletherapy in relation to conventional rehabilitation, (5) written in English. The review was conducted in accordance with the PRISMA statement. (Figure 1) [11]. The question generated using the Population, Intervention, Comparison, Outcomes, Study Design (PICOS) components was: “In post-stroke patients (P), is teletherapy (I) an effective rehabilitation tool for improving the functional condition (O) in reference to inpatient rehabilitation (C)?” Only randomised controlled trials (RCT) were included in the qualitative synthesis because they are considered as the best research design for evaluating the effectiveness of clinical interventions (S) [12].

The review conducted according to the PRISMA method yielded 10 papers, which are shown in Table 1.

## 3. Results

An analysis of the individual studies included in the review shows a high convergence in the inclusion and exclusion criteria for patients in the telerehabilitation group (Table 2). The average age of the participants was between 57 and 62 years old. Intervention sample sizes varied widely (e.g., in Cramer’s study *n* = 62, in Burgos’ study *n* = 6) [13,14].

A description of each publication, including information on the therapeutic interventions undertaken, the scales and tests used, and the results obtained, is provided below.

### 3.1. Impact of Virtual Reality (VR)-Based Telerehabilitation on the Functional Condition of Stroke Patients

In a randomised trial, Lisa Sheehy et al. evaluate the feasibility of virtual reality (VR)-based therapy at home as part of a telerehabilitation system for post-stroke patients [3]. In addition, qualitative and quantitative indicators of change in the functional status, ability to engage in the therapeutic programme, and the number and nature of adverse events are assessed. The therapy uses computer software that tracks the user’s movements and allows them to interact with a game presented on a television screen. Twenty patients, who had survived a stroke in the past 18 months, and were able to maintain a standing position for at least 2 min, without cognitive impairment, were randomly assigned to the experimental and control groups. Patients in the experimental group used VR to train balance, correct gait stereotype, and reach with the directly affected upper limb. The control group used an iPad in therapy with training apps affecting cognitive ability, fine motor skills of the hand, and visual tracking. Both groups were instructed in the therapeutic programme prior to beginning therapy. The intensity and difficulty of the VR tasks were monitored and adjusted remotely. Therapy lasted 30 min, 5 days a week, for 6 weeks. The therapy used the Jintronix Rehabilitation system, designed to incorporate motor control principles. Patients were provided with a motion tracking camera (asynchronous user monitoring) and software to eliminate the need for gloves/controllers, etc. The patients underwent the following tests before and after completing the rehabilitation programme: Berg Balance Scale (BBS), Timed Up-and-Go (TUG), 5 Times Sit-to-Stand (FTSST), Community Balance and Mobility Scale (CB&M), Stroke Impact Scale (SIS) (Table 3). Experimental group participants averaged 26.2 sessions, 27 min./session (77.8% compliance); control group participants averaged 33 sessions, 37 min./session (137.9% compliance) (*p* = 0.11 for sessions, *p* = 0.002 for the total duration). Mixed ANOVA showed no interaction between the group and time for any of the outcome measures. The only difference between groups concerned the FTSST test (*p* = 0.017). No falls or serious adverse events were reported [3,9,10].

### 3.2. Effect of Telerehabilitation on Functional Condition of the Occupied Upper Limb and Level of Stroke Knowledge

In a study by Steven C. Cramer et al., conducted at 11 centres in the United States, 124 patients with upper limb motor deficits (Fugl–Meyer (FM) score 22–56), who were 4–36 weeks post-stroke, without cognitive impairment, were studied [14]. The effect of home-based telerehabilitation (combined with an educational module) after 36 treatment sessions (70 min each for 6 weeks) on the functional status of the directly affected upper limb and stroke knowledge was evaluated. The control group (IC) received therapy in an inpatient setting. Patients in the experimental group (TR) completed 35.4 of 36 (98.3%) assigned treatment sessions, while patients in the IC group completed 33.6 of 36 (93.3%) sessions. The mean change in FM score in the TR group increased by 7.86 points from baseline (*p* < 001), in the IC group by 8.36 points (*p* < 001). The covariance-adjusted mean change in FM score was 0.06 points (95% CI, −2.14 to 2.26) higher in the TR group (*p* = 96). The equivalence margin amounted to 2.47 and was outside the 95% CI, proving the comparable effectiveness of rehabilitation in both systems. At screening, patients in the TR group correctly answered an average of 22.4 of 30 questions (74.7%) from the stroke knowledge exam, whereas in the IC group, the number of correct answers averaged 22.8 of 30 questions (76%). After completion of the treatment programme, the outcome improved by 3.3 = 11% correct responses (TR group) and by 2.5 = 8.3% (IC group) [14,17,18].

### 3.3. Impact of Social Networks on the Course and Effects of Telerehabilitation

In contrast, Podury A. et al. examined the impact of social networks on the course of home telerehabilitation and the association of specific social factors with improved functional status and reduced depressive symptoms [2,4]. Thirteen patients who had experienced a stroke 2 to 16 months prior, with hand motor deficits, underwent supervised teletherapy. Key inclusion criteria: age ≥ 18 years, motor deficits of the upper limb directly involved in FM-A: 28–66 (if FM-A > 59, a Box and Block score needed to be obtained on the hemiparesis side > 25%, compared to the indirectly affected side, functional status of the upper limb scoring ≥ 3 blocks in 60 s); exclusion criteria: active neurological or psychiatric comorbidity, major depression (Geriatric Depression Scale score > 11), significant cognitive impairment (Montreal Cognitive Function Assessment score < 22), communication deficits [2,19]. The therapy programme lasted 12 weeks (1 h per day, 6 days per week). It included routine assessment of upper and lower limb motor function of the directly affected limb and mood. At the midpoint of the telerehabilitation programme (week 6), the researchers mapped each study participant’s personal social network to assess the relationship between social network metrics and improvements in the functional status. For this purpose, the personal network analysis and quantitative assessment tool for social network structure and composition, PERSNET, was used [6,20]. The results were compared with a historical cohort of 176 post-stroke patients (who did not receive telerehabilitation) to determine the differences in social networks. We demonstrated a correlation between network size and density and improved gait time (*p* = 0.025; *p* = 0.003). Social network density was associated with improved upper limb motor skills (*p* = 0.003), while network size was associated with reduced depressive symptoms (*p* = 0.015). Telerehabilitation patient networks were larger (*p* = 0.012) and less dense (*p* = 0.046) relative to historical patient networks. Median FM-A improved significantly from baseline 46 to 59 (*p* = 0.0005), median gait speed improved from a score of 0.94 to 1.01 (*p* = 0.0007), Geriatric Depression Scale decreased from 3 to 1 points (*p* = 0.05) [2,21,22].

### 3.4. Effects of Tele Aerobic Training on Cardiorespiratory Fitness of Post-Stroke Patients

Researchers Galloway M. et al. evaluated the feasibility and satisfaction levels of home aerobic training of 21 post-stroke patients (≥3 months after stroke) [23]. The exercises were held 3 days a week and were supervised by telephone. Post-stroke adults living with caregivers, able to walk independently (FAC score ≥ 3), and without cognitive impairment were eligible for the study. Participants underwent an 8-week programme of moderate to intense intensity (55–85% of maximum heart rate as determined by the Borg Rating of Perceived Exertion (RPE) between 13 and 16). Patients were assigned to one of four groups that differed in the length of the therapy session (1 group: 10 min./session, 2nd group: 15 min./session, 3rd and 4th group: 20 min./session). Prior to the commencement of telerehabilitation, each patient received instructions demonstrating the correct execution of each task of the therapeutic programme. Exercises were progressive, individually tailored to the patient’s initial level of functional status, degree of disability, and movement preferences. Clinical assessment was used to modify motor tasks if the target heart rate was not reached or the patient experienced difficulty or discomfort. Cardiorespiratory fitness was measured by indirect spirometry during the first and last week of telerehabilitation during a 6-min walking test and a bicycle ergometer exercise test [24,25,26,27]. Patients’ satisfaction with telerehabilitation was assessed using a questionnaire (23 multiple choice + 2 open-ended questions) at the end of the study. It was shown that 95% of the participants would undergo rehabilitation again in the tele system. On the one hand, the exercises turned out to be demanding enough to improve the efficiency and functional status of the patients; on the other hand, they were safe and convenient—the patients appreciated the therapy at home, which did not generate, for example, problems related to transportation [23,28,29,30,31].

### 3.5. Assessing the Impact of Telerehabilitation Based on a Collaborative Model on the Functional Status of Post-Stroke Patients

Chinese researchers have examined the effectiveness of a telerehabilitation programme for the purpose of improving post-stroke patients in the acute phase, based on a collaborative model [32]. Sixty-one subjects were studied (30 cases—intervention group, 31 cases—control group). Exclusion criteria included cognitive and psychiatric impairment, comorbidities affecting motor function, complete aphasia, and severe visual impairment. During hospitalisation, patients in the control group received routine early rehabilitation instructions. The content addressed physiological motor patterns of the upper and lower limbs, transfer from the lying position to a sitting one, or maintaining a proper range of motion in the joints. After hospital discharge, patients received rehabilitation and medication counselling by telephone (once a week). Patients in the intervention group received identical instructions during hospitalisation pertaining to early rehabilitation, but were provided remote home rehabilitation based on a collaborative model after hospital discharge. A care team consisting of neurologists, physiotherapists, nurses, counsellors, and caregivers was formed for this purpose. Remote in-home rehabilitation delivery was based on the TCMeeting v6.0 web-based videoconferencing system consisting of a computer, projector, camera, and data archiving software. The functional status of the patients was described by the Fugl–Meyer test, Berg Balance Scale, Up and Go Test, and a 6-Minute Walking Test [32,33]. In addition, the Stroke Specific Quality of Life Scale (SSQoL) was used to assess the subjects’ ability to perform daily living activities. Changes in functional condition and motor control ability were assessed at 4, 8, and 12 weeks after inclusion in the study (Table 4).

### 3.6. Effect Of Telerehabilitation on Improving Cognitive Function in Post-Stroke Patients

Torrisi M. et al. examined the effect of telerehabilitation on improving cognitive function in post-stroke patients [34]. Forty patients were studied for this purpose (20 patients were assigned to the control group (CG), 20 to the experimental group (EG)). In the first phase of the programme, both groups underwent inpatient rehabilitation training: EG patients received cognitive training using VRRS-Evo (3D scenarios), CG patients underwent identical pencil and paper exercises. In the second phase of the programme (after hospital discharge), the EG group continued cognitive function therapy using a home VRR tablet (2D scenarios), whereas the CG group received traditional therapy (3 sessions per week, each session approximately 50 min.). The efficacy of telerehabilitation for the treatment of cognitive disorders was demonstrated, particularly improvements in global cognitive function levels, as well as in attentional, memory and language skills in EG [34,35].

### 3.7. Level of Acceptance of Telerehabilitation in Stroke Patients

Chen Yu et al. performed a qualitative study on the acceptance of home-based telerehabilitation by conducting in-depth interviews with 13 post-stroke patients (4–36 weeks after stroke), who completed 6 weeks of teletherapy [15]. Inclusion criteria included: age ≥ 18 years, FM-A score between 22–56, Box and Block test score of upper limb directly involved > 3 blocks in 60 s, no cognitive impairment. The telerehabilitation system consisted of four main components: games, exercise, stroke education, and telecommunications. The therapy programme consisted of 70-min sessions, 6 days a week, for 6 weeks. User acceptance of the telerehabilitation system was analysed using the Unified Theory of Acceptance and Use of Technology (UTAUT), a model of technology acceptance and use that describes four factors: expected outcomes, expected effort, facilitating conditions, and social impact [36]. The qualitative results of the study revealed high levels of participant satisfaction with teletherapy. Patients reported improvements not only in their functional status, but also in mood, cognitive ability, or social interaction. Telerehabilitation, on the one hand, provided repetitive tasks and, on the other hand, was characterised by a diversity of exercises. The system offered external (physiotherapist monitoring of outcomes) and internal (patient influenced progress in therapy) motivation to engage in the therapy programme. The flexibility of the time and location of the tele system sessions was an additional advantage, as it offset transportation issues. The only limitation, according to study participants, was barriers in technical/technological skills, but problems of this nature were resolved by contacting the research team [15].

### 3.8. Effect of Telerehabilitation on Balance Improvement in Post-Stroke Patients

Burgos P. et al. conducted a study evaluating the effect of telerehabilitation on balance improvement in post-stroke patients [13]. The tele system was based on games installed on smartphones. Additionally, participants were provided with inertial motion sensors (IMUs) and cloud databases. The research involved six patients 6–8 weeks after stroke. Inclusion criterion: BBS score < 50, availability of caregiver during the patient’s rehabilitation programme. Therapy lasted 4 weeks (a single session lasted 30 min.). The control group consisted of four people. Telerehabilitation was preceded by training the participants and their caregivers on safety and how to use the equipment. Balance was assessed with the Berg balance scale at the beginning and end of the rehabilitation programme [37,38]. In addition, the System Usability Scale (SUS) was used to evaluate the user experience. The scale consists of 10 questions relating to frequency and difficulty of use, system complexity, safety, or study participants’ prior knowledge of the technology system. BBS scores improved with mean values: PRE = 35 ± 4.42 (62.5% ± 7.91), POST = 46.33 ± 3.01 (82.67% ± 5.37). Compared with the control group, the BBS PRE-POST variance in the study group was higher at 20.2% ± 6.36 vs. 12.5% ± 8.63, with the difference in variance between groups being statistically significant (*p* = 0.019). The mean SUS score was greater than 80 (87.5 ± 11.61), illustrating an excellent level of usability of the system for use [13,39,40,41].

### 3.9. Feasibility of a Tele Therapeutic Program in Post-Stroke Patients

Simpson D. et al. evaluated the degree of task feasibility in telerehabilitation, as well as the ability to verify exercise accuracy and functional progress in a tele system [42]. The study group consisted of 10 stroke survivors. Therapy lasted 4 weeks and included sitting and standing exercises [43]. The therapist remotely monitored adherence to movement commands and progress toward goals, and provided feedback to patients via the app. Inclusion criteria: age ≥ 18 years, stroke within the past two years, ability to stand up independently from a seated position from a chair. Each participant was instructed on proper task performance, safety and proper use of the technology. At baseline and at the end of the treatment programme, each patient completed a 2-min STS test and completed the System Usability Scale. Functional status was described using the Short Set of Physical Performance Battery (SPPB), which measures three aspects of physical status: gait speed, balance, and muscle strength. SPPB is a composite score from 0 to 12, where the higher the score, the better the patient’s functional status. The change in scores from 0.99 to 1.34 points was considered clinically significant for older adults. During the study, participants performed a total of 224 exercise sessions out of the recommended 184 sessions. Individually, patients performed an average of 750 repetitions from sitting to standing (range 385–1410) over the course of 4 weeks, compared with an average of 724 prescribed repetitions (range 398–1395). The therapist made progress on the average goal from week 1 to week 4 by 92%. Participants rated the system usability at 79% [42,44,45].

### 3.10. The Impact of Telerehabilitation on the Functional Condition of Stroke Survivors in African Countries

Nigerian researchers, on the other hand, have developed a tele home exercise programme for post-stroke speakers of indigenous African languages [46]. The video-based program (VHEP) was conducted in the Yoruba language. In the course of its development, recommendations from the American Stroke Association were followed to include instruction in task-specific motor and postural exercises, trunk exercises, and correct gait stereotype exercises. Inclusion criteria: history of stroke three or more months ago, modified Ashworth Scale score ≥ 1, Brunnstrom score ≥ 3, no mental illness, epilepsy, cancer, heart disease. The research involved 10 people. The feasibility questionnaire was adapted from the Satisfaction Survey for the Individual with Stroke form from a smartphone educational intervention study. Each motor task was demonstrated on video for 5 min. The videos began with an introduction, followed by 5 short exercises in various positions (lying, kneeling, sitting, standing, walking). The main focus of the therapy was repetition, a gradual increase in task complexity, and functional training at a self-selected pace. The total time for a single exercise intervention was 30 min. Patients’ exertion magnitude was monitored throughout the treatment programme based on feedback from the Modified Borg Perceived Exertion Scale (RPE). The study proved that the VHEP system was goal-oriented and had a high level of acceptability for motor tasks. With a demonstration of each exercise, therapy proved easy and safe [18,46,47,48]. As an added bonus, the Yoruba language also made the tool accessible to patients who speak indigenous African languages [49].

Table 5 summarises publications based on randomised controlled trials (RCT) and their characteristics according to PRISMA methodology.

## 4. Discussion

This review examines the efficacy results of teletherapy on various aspects of the functional status of post-stroke patients. Research has shown that the tele system is a potential solution to the problem of increasing demand for rehabilitation services in a resource-constrained environment. Telerehabilitation provides continuity and an appropriate level of therapy intensity by using increased repetition of motor tasks [7,50]. Clinical research indicates that hundreds of repetitions in a specific movement pattern are required to achieve an optimal range of motor cortex neuroplasticity after stroke. Cramer’s study calculated 1031 repetitions of upper limb movements per day in the intervention trial, demonstrating the effect of teletherapy in maximising the plasticity phenomenon by intensifying the applied therapy (the number of repetitions of upper limb movements during conventional therapy averages 32 per session) [14]. Telerehabilitation therefore contributes to a significant increase in function, regardless of whether therapy was initiated in the acute phase (<90 days from the stroke incident) or the chronic phase (>90 days after stroke). At the same time, it alleviates the problem of patient transport, which is cited as a major limitation in access to inpatient therapy, and reduces the expenses incurred for post-stroke rehabilitation in the private sector [34]. In turn, direct contact with a therapist, in the case of in-centre rehabilitation, plays an important role in the improvement process, increasing the patient’s sense of security and support as well as the correctness of patients’ motor tasks [42]. Despite this, in certain cases, rehabilitation in the tele system has comparable efficacy to therapy provided in a traditional inpatient environment, as evidenced by the results of the aforementioned review (Figure 2). Patients undergoing teletherapy are characterised by high commitment to the goals of the improvement programme (optimisation on the primary and secondary control scales) and satisfaction with rehabilitation outcomes (positive change in the Physical Activity Satisfaction Scale) [42]. The effectiveness of post-stroke therapy is closely related to the patient’s motivation; however, maintaining it (motivation) at an appropriate level is challenging. The rate of non-adherence to therapy recommendations can reach 70%, especially in the case of home exercises without direct supervision by a physiotherapist [14]. A motivated patient, on the other hand, experiences less resistance to the rehabilitation programme [51]. The flexibility of therapy location and time, guaranteed by the tele system, minimises the burden imposed by traditional inpatient sessions. As a result, teletherapy participants acknowledge that teletherapy improvement provides the discipline necessary for regular exercise and contributes to high adherence among patients. In addition, they perceive benefits relating to both physical (improved performance, function, sleep) and psychological (increased confidence, motivation to achieve goals) status, which amounts to increased quality of life and reduced risk of secondary stroke [23]. Patients also often experience shrinkage of social networks after strokes. This is mainly due to loss of contact with friends, decreased participation in group events or avoidance of social activities [2,16]. This phenomenon can contribute to worsening disability, as social isolation worsens indicators of functional status and increases the risk of depression [6]. The interaction with the therapist, despite taking place in the form of video conversation, on the one hand ensures that the exercises are performed correctly and that the results are analysed, and, on the other hand, it improves the patient’s mood and makes him or her feel less socially isolated. The Podury’s study shows that Geriatric Depression Score significantly decreased over 12-weeks supervised home-based telerehabilitation from 3 (1–5) to 1 (0–4) (*p* = 0.05) and it also found that patients undergoing telerehabilitation had larger and more open social networks than those in the control group [2,21,22]. A correlation was observed between the size and density of the network and the improvement in patients’ gait time. Social network density was also associated with improvements in upper limb motor skills, while network size was associated with reductions in depressive symptoms. Furthermore, telerehabilitation using virtual reality (VR) is characterised by a large variety of goal-oriented motor tasks due to the availability of a wide range of interactive games with progressive levels of difficulty. The motor experiences offered by VR make traditional exercise unattractive to patients. In the Sheehy study, the tele technologies were proven to be easy to use and affordable for the stroke population eligible for this type of therapy [3]. Study participants came from a wide age range and had varying levels of disability or familiarity with technology, which was not a barrier to successful participation in telerehabilitation sessions. The Chen research shows that teletherapy can be used as an efficient and user-friendly tool to deliver home-based stroke rehabilitation that enhance patients’ physical recovery and mental and social-emotional wellbeing. Participants mostly reported positive experiences with the tele system. Benefits included observed improvements in limb functions, cognitive abilities, and emotional well-being. They also perceived the system to be easy to use due to the engaging experience and the convenience of conducting sessions at home [15,16,17,18,19,20,33,52,53]. Only in the publications by Cramer and Galloway were there cases of dropouts during research (Cramer—10 patients, Galloway—3 patients). The reason for this, however, was not difficulties arising from the use of the tele system but personal reasons or the need to return to work during the rehabilitation program [14,23]. Teletherapy has also had a tremendous impact on increasing the level of knowledge of strokes and secondary prevention, which are often inadequate among post-stroke patients. Through educational programmes that complement the teletherapeutic process, an optimisation of health is achieved, resulting in improved functional status and reducing the risk of secondary stroke. Thus, patient education is one of the key components of effective telerehabilitation [14]. Unfortunately, the strength of the evidence itself supporting the effectiveness of teletherapy still ranks low. The reason for this is the limited number of randomised controlled trials, small number of study participants, or heterogeneous trials [12]. Participants in the tele-improvement programme are primarily patients with mild to moderate motor disabilities who qualify for therapy based solely on specifically defined criteria (Table 2). Teletherapy itself obligatorily has to be preceded by understandable instructions and must be constantly controlled and modified by a physiotherapist. To ensure safe exercise progress, the entire improvement programme takes place in the presence of a supervisor. Nevertheless, based on the above-mentioned literature review and considering the lack of sufficient RCTs, it can be concluded that telerehabilitation is an alternative modality to conventional therapy, facilitating post-stroke patients’ achievement of functional recovery and an improvement in their quality of life in the resource-limited settings caused by the SARS-CoV-2 pandemic.

## Figures and Tables

**Figure 1 healthcare-09-00654-f001:**
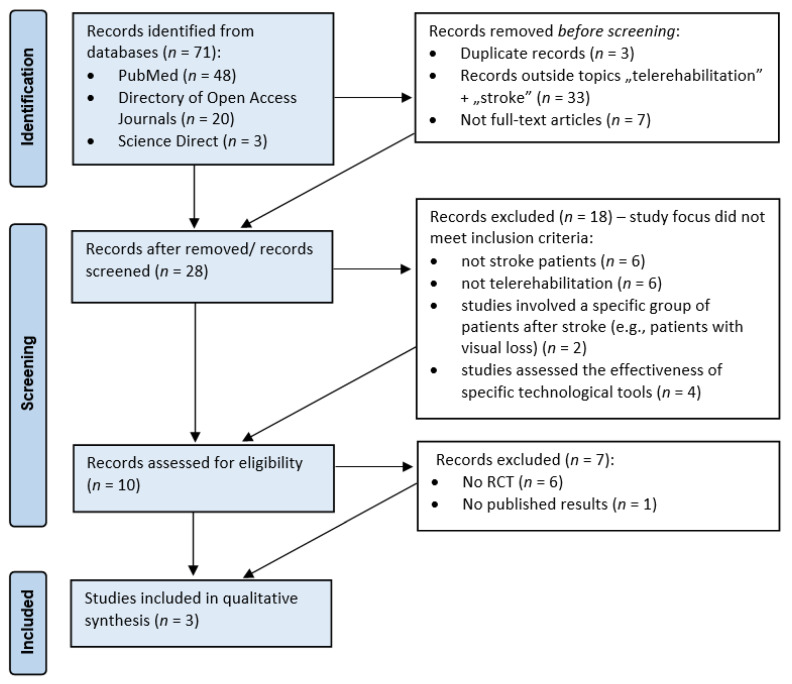
Flow diagram adapted from PRISMA which shows the process for identifying and screening the articles for inclusion and exclusion.

**Figure 2 healthcare-09-00654-f002:**
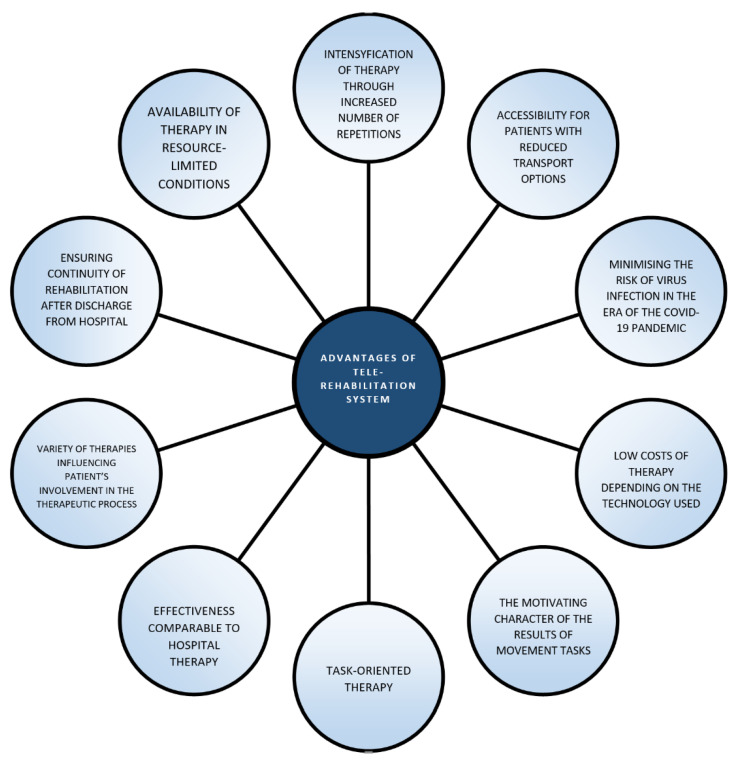
Advantages of telerehabilitation.

**Table 1 healthcare-09-00654-t001:** Description of articles initially included by PRISMA methodology.

Article Type	Focus	Reference
RCT with published results	Home-based TR for adults after stroke Collaborative care model based TR for acute stroke patients TR to improve balance in stroke patients	Cramer S. et al., 2019 Wu Z. et al., 2020 Burgos P. et al., 2020
RCT without published results	Home-based VR rehabilitation after stroke	Sheehy L. et al., 2019
Case study	User acceptance of a home-based stroke TR Video home-based TR for stroke survivors	Chen Y. et al., 2020 Odetunde M. et al., 2020
Other trials (not randomized and/ or not controlled)	Correlation of social network structure and home-based TR after stroke TR aimed at increasing cardiorespiratory fitness for people after stroke TR using to improve cognitive function in post-stroke survivors Connecting stroke survivors and therapists using technology	Podury A. et al., 2021 Galloway M. et al., 2019 Torrisi M. et al., 2019 Simpson D. et al., 2020

**Table 2 healthcare-09-00654-t002:** Inclusion and exclusion criteria for patients enrolled in the study program [15].

**Inclusion Criteria:** Age ≥ 18 years Past stroke (usually with time of stroke onset 4–36 weeks prior to study program) Box and Block Test results of directly involved upper limb ≥ 3 blocks in 60 s FM-A score of 22–56/ 66 points Ability to transfer independently from sitting to standing The presence of a caregiver while doing movement tasks
**Exclusion Criteria:** Diagnosis (apart from the index stroke) significantly affecting the function of the upper or lower limb directly involved A major, active, coexistent neurological or psychiatric disease (including dementia) A medical disorder that substantially reduces the likelihood that a subject will be able to comply with all study procedures Complete aphasia Severe depression, defined as Geriatric Depression Scale Score > 10/15 Significant cognitive impairment, defined as Montreal Cognitive Assessment score < 22/30 Deficits in communication that interfere with reasonable study participation Lacking visual acuity, with or without corrective lens, of 20/40 or better in at least one eye Receipt of Botox to arms, legs or trunk in the preceding 6 months, or expectation that Botox will be administered to the arm, leg or trunk prior to study program Unable or unwilling to perform study procedures/ therapy, or expectation of noncompliance with study procedures/ therapy Pregnancy

**Table 3 healthcare-09-00654-t003:** Scores of scales and functional tests before-post telerehabilitation [3,16].

Post-Pre Assessment Average (95% Confidence Intervals):	Experimental Group (EG)	Control Group (CG)	*p*-Values for Change Over Time (EG & CG Combined)
TUG (seconds)	−0.1 (−1.8, 1.6)	−1.4 (−3.7, 1)	0.326
TUG + cognitive task (seconds)	−1.7 (−4.2, 0.7)	−3.4 (−5.7, −1.1)	0.004
FTSST (seconds)	−3 (−5.8, −0.2)	−2 (−4.1, 0.2)	0.006
BBS (/56)	−0.5 (−4.2, 3.3)	0.6 (−0,5, 1.6)	0.959
CB&M (/96)	5.6 (−5, 16.2)	6.1 (1.6, 10.7)	0.049
SIS (/295)	7.7 (−2.1, 17.6)	13.8 (2.2, 25.3)	0.006

**Table 4 healthcare-09-00654-t004:** Changes in scores of functional condition and motor control abilities after 4, 8 and 12 weeks of telerehabilitation [32].

	Intervention Group				Control Group				RM-ANOVA	
	Baseline	4 Weeks	8 Weeks	12 Weeks	Baseline	4 Weeks	8 Weeks	12 Weeks	F (Time*Group)	*p*
FM(UE)	11.93 ± 2.50	35.90 ± 2.78	49.10 ± 3.00	55.33 ± 2.81	2.61 ± 1.78	29.35 ± 2.36	39.35 ± 4.13	47.42 ± 3.90	42.523	<0.001
FM(LE)	13.37 ± 1.38	23.87 ± 1.28	25.50 ± 1.74	28.37 ± 2.51	14.13 ± 1.43	20.84 ± 1.39	24.23 ± 1.86	27.87 ± 1.73	57.000	<0.001
BBS	21.07 ± 3.29	30.50 ± 2.84	38.13 ± 2.84	43.13 ± 2.32	20.87 ± 2.33	28.06 ± 2.28	34.19 ± 2.15	38.29 ± 2.70	9.205	<0.001
TUG	41.93 ± 3.57	30.37 ± 3.62	22.73 ± 2.49	19.50 ± 2.73	40.58 ± 4.40	34.23 ± 2.86	27.13 ± 2.50	23.97 ± 3.35	16.320	<0.001
6MWT	91.73 ± 7.46	111.50 ± 8.12	128.90 ± 7.42	141.63 ± 8.68	92.35 ± 6.15	107.94 ± 5.14	123.13 ± 5.71	129.45 ± 7.06	10.530	<0.001

Note: F—time effect group (time factor * grouping factor).

**Table 5 healthcare-09-00654-t005:** Description of RCT included by PRISMA methodology in this review.

Authors/Year	Participants	Intervention	Outcomes Measurement	Results
Cramer S. et al., 2019	*n* = 124 (34 women, 90 men) (*n* = 62 in experimental group) Mean age (SD) of 61 years Adults ischemic stroke or intracerebral hemorrhage 4 to 36 weeks prior	Experimental and control group received 18 supervised and 18 unsupervised 70-min sessions. The treatment approach was based on an upper-extremity task-specific training manual and Accelerated Skill Acquisition Program All sessions for both groups included at least 15 min per day of arm exercises (the same 88 exercises for both groups) and at least 15 min per day of functional training. The TR system (for experimental group) consisted of an internet-enabled computer with table, chair, and 12 gaming input devices, but no keyboard, as no computer operation was required by patients.	FMA-UE (Fugl-Meyer Assessment upper extremity) Box and Block Test SIS (Stroke Impact Scale)	Both groups showed significant treatment-related motor gains, with a mean (SD) unadjusted FM score change from baseline to 30 days after therapy of 8.36 (7.04) points in the control group (*p* < 001) and 7.86 (6.68) points in the experimental group (*p* < 001). The adjusted mean change in FM score was 0.06 points larger in the experimental group (95% CI, −2.14 to 2.26; *p* = 96). The noninferiority margin (30% of the mean FM score change in the control group) was 2.47, which fell outside of this 95% CI, indicating that TR was not inferior to standard therapy on the primary end point [14]. Box and Block Test scores increased by 9.5 (*p* < 001) in the experimental group and by 8.8 (*p* < 001) in the control group and indicated noninferiority of TR therapy. Stroke Impact Scale hand motor domain scores increased by 23.7 (*p* < 001) in the experimental group and by 29.2 (*p* < 001) in the control group, although noninferiority was not demonstrated with this outcome [14].
Wu Z. et al., 2020	*n* = 61 (25 women, 36 men) (*n* = 30 in experimental group) Mean age (SD) of 58 years Adults ischemic or hemorrhagic stroke No information about time since stroke	Patients in the intervention group received home remote rehabilitation based on a collaborative care model. A collaborative care team consisting of neurologists, nurses, rehabilitation therapists, counselors and caregivers was established. Rehabilitation therapists assess the extent of patient dysfunction and work with family caregivers to develop rehabilitation plans and goals. The home remote rehabilitation guidance uses the Internet-based TCMeeting v6.0 video conferencing system. Patients in the control group received only routine rehabilitation and nursing measures, including dietary guidance, medication guidance and rehabilitation guidance, which were conducted by telephone follow-up once a week.	FMA-total (Fugl-Meyer Assessment total) FMA-UE (Fugl-Meyer Assessment upper extremity) FMA-LE (Fugl-Meyer Assessment lower extremity) BBS (Berg Balance Scale) TUG (Timed “Up&Go” Test) 6 MWT (6-min Walk Test) MBI (Modified Barthel Index)	See Table 4.
Burgos P. et al., 2020	*n* = 10 (4 women, 6 men) (*n* = 6 in experimental group) Mean age (SD) of 61 years. Adults ischemic or hemorrhagic stroke in subacute (6–8 weeks after stroke)	Both groups received their standard rehabilitation treatment at the hospital site (3 sessions of 40 min per week of physical therapy for 4 weeks). In addition, the intervention group received 9 sessions of 30 min per week for 4 weeks. In each session, participants trained in balance tasks using smartphone-based exergames controlled by body motions.	BBS (Berg Balance Scale). MBT (Mini-BESTest). BI (Barthel Index) SUS (System Usability Scale)	Balance results improved in the BBS, with mean values of PRE = 35 ± 4.42 (62.50% ± 7.91), POST = 46.33 ± 3.01 (82.67% ± 5.37), and MBT PRE = 10.33 ± 2.87 (36.89% ± 10.26), and POST = 18.67 ± 2.81 (66.67% ± 10.01) with a statistically significant variation within PRE and POST (F(1/5) = 60.84, *p* < 0.001 and F(1/5) = 45.96, *p* = 0.001, respectively). Functional independence, measured by BI, also improved in the study group with PRE = 65.00 ± 4.47, and POST = 82.50 ± 8.80. There was also a statistically significant variation within PRE and POST times (F(1/5) = 18.85, *p* = 0.007) [13]. In comparison with the control participants, BBS variation PRE–POST for the study group was higher, with 20.20% ± 6.36 vs. 12.50% ± 8.63, with a statistically significant difference in the variation between groups (F(1/7) = 9.15, *p* = 0.019; Cohen-d = 2.98). For MBT PRE–POST variation, it was 29.7% ± 10.75 in the telerehabilitation group and 16.96% ± 9.39 in the control group, without significant differences between groups (F(1/7) = 1.61, *p* = 0.245; Cohen-d = 2.94). Functional independence (BI) in participants trained with our telerehabilitation system was higher compared to the controls: 17.50 ± 9.87 vs. 3.75 ± 8.53, with a significant difference between groups (F(1/7) = 7.97, *p* = 0.025; Cohen-d = 2.50) [13]. The average SUS score was higher than 80 (87.5 ± 11.61), which can be interpreted as an excellent system user usability level [13].

## Data Availability

The data presented in this study are openly available in PubMed, Directory of Open Access Journals and Science Direct databases.

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
