# Peer review of "Telerehabilitation of Post-Stroke Patients as a Therapeutic Solution in the Era of the Covid-19 Pandemic"

_healthcare, 2021, doi:10.3390/healthcare9060654_

Round 1
Reviewer 1 Report
Authors had better add the patient profile about how long days or months after onset of the index stroke.
Authors described clearly which measurements were significant differences between control and experimental groups in each study. There were no p-values in table 3 and 4.
Author Response
Response to Reviewer 1 Comments
Thank you very much for your essential comments regarding incomplete data in patient profiles and test results. The missing information has been completed accordingly:
Point 1: Authors had better add the patient profile about how long days or months after onset of the index stroke.
Response 1: In lines 122, 170-171, 197, 265-266, 289-290 we have added information about the period since the stroke incident.
Point 2: Authors described clearly which measurements were significant differences between control and experimental groups in each study. There were no p-values in table 3 and 4.
Response 2: In lines 144, 246 we have inserted new tables with missing data.
Thank you again for your remarks.
Reviewer 2 Report
Nicely written and adequately supported manuscript regarding telerehabilitation in the era of covid-19 pandemic.
However, there are some points that deserve special mention from my side and I wish you could provide me more details or clarifications. More specifically:
You mention that ‘This is a period of increased neuroplasticity in which major reorganizational changes occur in the brain.’ I think that I understand what you mean under the term ‘neuroplasticity’. But, if we want to use the precise definition of this term, we should only use it for children up to about 6 years of age, especially regarding language function. I recommend that you could better define and explain what you mean under this term, especially if you are referring to adult patients.
You state that ‘Social isolation, which negatively affects the functional and emotional state of post-stroke patients, is an individual risk factor for stroke recurrence.’ I would be very reluctant to state that social isolation is an individual risk factor for stroke recurrence based on only a single literature report, as it appears from your manuscript.
At another point you refer that ‘Selection was based on inclusion criteria: written in English.’ I was wondering why articles not written in English should be excluded from your study, if they were included in your research based on other inclusion criteria.
Finally, you mention that ‘in certain cases, rehabilitation in the tele system has comparable or even greater efficacy to therapy provided in a traditional inpatient environment’. I am receptive to agree that rehabilitation in the tele system has comparable efficacy to therapy provided in a traditional inpatient environment but I am hesitant to accept that it’s efficacy could be greater.
Author Response
Response to Reviewer 2 Comments
Thank you very much for your essential comments.
Point 1: You mention that ‘This is a period of increased neuroplasticity in which major reorganizational changes occur in the brain.’ I think that I understand what you mean under the term ‘neuroplasticity’. But, if we want to use the precise definition of this term, we should only use it for children up to about 6 years of age, especially regarding language function. I recommend that you could better define and explain what you mean under this term, especially if you are referring to adult patients.
Response 1: Of course, we need to clarify the definition of plasticity as the term "neuroplasticity" itself is too general. We meant post-damage (compensatory) plasticity, which we noted in line 44: "post-damage (compensatory) plasticity". Thank you again for your remark. Indeed, it would have been a serious mistake to leave a general definition of neuroplasticity.
Point 2: You state that ‘Social isolation, which negatively affects the functional and emotional state of post-stroke patients, is an individual risk factor for stroke recurrence.’ I would be very reluctant to state that social isolation is an individual risk factor for stroke recurrence based on only a single literature report, as it appears from your manuscript.
Response 2: Regarding the second remark, we agree with a too general statement, based on a single article, that social isolation is an individual risk factor for stroke recurrence. In lines 50-53 we have noted that this is a conclusion derived from a specific publication, and that this statement cannot be applied to the entire population of post-stroke patients ("Dhand's research shows that social isolation, which negatively affects the functional and emotional state of post-stroke patients, is an individual risk factor for stroke recurrence. However, this single report should not be valid with respect to the entire population of stroke survivors.").
Point 3: At another point you refer that ‘Selection was based on inclusion criteria: written in English.’ I was wondering why articles not written in English should be excluded from your study, if they were included in your research based on other inclusion criteria.
Response 3: While reading the review, we noticed how unfortunate it was to use the exclusion criterion base on the English language. We intended to narrow down the articles written in a specific language, not to English-speaking countries. We would only like to note that, fortunately, in our work we also quoted articles from China or African countries, and the research included in these publications embraced patients from the countries the authors of the articles originated from. Next time we will definitely make sure not to select such a criterion because it may introduce unnecessary restrictions. Once again, we are grateful for your important comments thanks to which we will not make a similar mistake in the future.
Point 4: Finally, you mention that ‘in certain cases, rehabilitation in the tele system has comparable or even greater efficacy to therapy provided in a traditional inpatient environment’. I am receptive to agree that rehabilitation in the tele system has comparable efficacy to therapy provided in a traditional inpatient environment but I am hesitant to accept that it’s efficacy could be greater.
Response 4: Regarding the last comment on greater effectiveness of telerehabilitation versus rehabilitation in the traditional environment, we have indeed made an excessive conclusion, perhaps derived from the desire to emphasize the importance of telerehabilitation, but of course not in the sense that it is more valuable than rehabilitation in direct patient-therapist contact. The content presenting greater effectiveness of telerehabilitation has been removed from the manuscript.
Thank you again for your remarks.
Reviewer 3 Report
Dear Editor,
Dear Author,
I did like to read the presented paper.
It presents a Meta-Analysis from publications which addresses an investigation for post-stroke rehabilitation in while the shortage of facilities in the COVID-19 pandemic in a tele-digitalized setting.
First of all the authors selected only studies conducted in a randomized design, which is important as selecting the patients without randomization will influence the results. Furthermore, the authors summarized the results precisely, presenting the differences in scales appropriately. The discussion is well balance addressing all arguments for or against tele-rehabilitation. The results shows the procedure represents a good alternative for rehabilitation after stroke, but the appropriate patients need to be selected. However, I have some formal points and some with respect to the content which could be considered.
- To increase the readability it would insert a table with the abbreviations; there are many. For example in table 1 you use TR. I could not find the explanation. Probably tele-rehabilitation is meant.
- In studies conducted in this environment the rates of dropouts are also important when interpreting the results. Could the authors add information on that?
- In the discussion the topic of “mood” is well discussed. Are data regarding mood in the presented studies reported? If yes, could this be added to the current analysis?
- Line 414: The wording “minimizing” could be changed into reducing. Minimizing sounds like reducing the risk for stroke to a lowest level. In the context of educational measures, minimising sounds exaggerated.
Author Response
Response to Reviewer 3 Comments
Thank you very much for your essential comments.
Point 1: To increase the readability it would insert a table with the abbreviations; there are many. For example in table 1 you use TR. I could not find the explanation. Probably tele-rehabilitation is meant.
Response 1: We apologize for such an oversight. We agree that all abbreviations used in the manuscript should be explained, therefore, as suggested in line 66, we have inserted a table of abbreviations used in the work.
Point 2: In studies conducted in this environment the rates of dropouts are also important when interpreting the results. Could the authors add information on that?
Response 2: Referring to the second item in lines 420-424, we have included information on the number of people dropping out of the study.
Point 3: In the discussion the topic of “mood” is well discussed. Are data regarding mood in the presented studies reported? If yes, could this be added to the current analysis?
Response 3: According to the third item, in lines 400-402 and 414-420 we have included specific figures and information on the positive impact of telerehabilitation on the patient's mood.
Point 4: Line 414: The wording “minimizing” could be changed into reducing. Minimizing sounds like reducing the risk for stroke to a lowest level. In the context of educational measures, minimising sounds exaggerated.
Response 4: We agree, of course, as to changing the word in the context of the sentence. The word "minimizing" sounded excessive, so we changed it to the suggested "reducing".
Thank you again for all your comments.
Round 2
Reviewer 2 Report
Dear Authors,
thank you for reconsidering your article submission based on my comments.
Indeed, I strongly consider that you have replied to my remarks and the quality of your article has been strengthened. In that form, I suggest that your revised manuscript could be accepted for publication.